# Three-dimensional analysis using a dental model scanner: Morphological changes of occlusal appliances used for sleep bruxism under dry and wet conditions

Aya Ozawa[1], Yoshitaka Suzuki[1]*, Kazuo Okura[1‡], Toshinori Okawa[2‡], Susumu Abe[2‡], Kohei Kamoi[3‡], Emi Uyama[4‡], Kenichi Hamada[4‡], Yoshizo Matuka[1‡]

1 Department of Stomatognathic Function and Occlusal Reconstruction, Graduate School of Biomedical Sciences, Tokushima University, Tokushima, Japan, 2 Department of Comprehensive Dentistry, Tokushima University Graduate School, Tokushima, Japan, 3 Department of Medical Technology, Division of Dental Technology, The University of Tokushima Hospital, Tokushima, Japan, 4 Department of Biomaterials and Bioengineering, Graduate School of Biomedical Sciences, Tokushima University, Tokushima, Japan

☯ These authors contributed equally to this work.
‡ These authors also contributed equally to this work.
* yosuzuki@tokushima-u.ac.jp

## Abstract

### Objective

This study used a dental model scanner and best-fit alignment to analyze the deformation patterns of stabilized occlusal appliances (OcAs) used to treat sleep bruxism when stored in wet or dry conditions.

### Methods

Eight OcAs were prepared using polymethyl methacrylate, stored in water at room temperature for 4 weeks (wet storage), and then in air for 4 weeks (dry storage). After being stored in water for one month for hydration, they were 3D-scanned and digitized on days 0, 28, and 56. 3D-deformation patterns were obtained by comparing the storage-D group data using a best-fit alignment program that aligns the nearest points of the corresponding images in a virtual space and calculates the difference between the two images. The maximum deviation and deformed area in the ± direction, and the volume of the wet (W) and the dry (D) condition groups were compared using Wilcoxon's signed rank test.

### Results

OcA showed no obvious deformities in the W group at 4 weeks. However, in the D Group, a typical deformation pattern was detected at 4 weeks, with the posterior margin of the molars shrinking anteriorly, the palatal side of the molars lifted, and the buccal side retracted inward. The D group was significantly larger than the W groups in terms of maximum deviation in the ± direction, deformed area, and volume.

**Data Availability Statement:** All relevant data are within the manuscript and its Supporting Information files.

**Funding:** The author(s) received no specific funding for this work.

**Competing interests:** The authors have declared that no competing interests exist.

## Conclusions

After 4 weeks of storage under dry conditions, OcA showed a typical deformation pattern of shrinkage toward the center, with in-center lifting toward the palatal side. Significantly greater surface deviations, deformation areas, and deformation volumes were observed under dry conditions than in wet conditions.

## Introduction

Tooth grinding and jaw clenching during sleep, commonly referred to as sleep bruxism (SB), are risk factors for various pathological conditions such as tooth wear, breakage, or removal of prostheses [1], exacerbation of periodontal disease, and temporomandibular joint (TMJ) disorders [2]. In SB, forces greater than the maximum voluntary bite force may be exerted [3], and the effect of this excessive bite force on oral and maxillary functions is the cause of the abovementioned symptoms. SB is often managed using a removable intraoral appliance known as a stabilizing-type occlusal appliance (OcA) that covers the occlusal surfaces of the dentition. OcA prevents direct loading, affects teeth, and distributes excessive occlusal forces by providing occlusal contact throughout the dentition. It also reduces the burden on the TMJ and dentition and protects the stomatognathic system from damage, such as a reduction in occlusal height diameter caused by pathological tooth attrition and TMJ disorders caused by the resulting burden on the TMJ [4]. There are also reports that SB is suppressed for only two weeks using OcA [5].

OcAs are often fabricated using poly (methyl methacrylate PMMA). After polymerization, PMMA undergoes dimensional changes due to expansion due to water absorption [6,7] and shrinkage during drying [8]. Sweeney [9] reported that because PMMA denture bases are affected by water absorption, the occlusal adjustment should be delayed until the PMMA is saturated with water. Therefore, to prevent their deformation, intraoral devices containing PMMA must be stored underwater.

Bohnenkamp [10] placed die pins on OcAs fabricated with PMMA, measured the pin-to-pin distance, and reported that storage under dry conditions for two weeks resulted in significantly shorter distances than storage under wet conditions. Lim and Lee [11] reported the three-dimensional (3D) deformation of complete dentures made of PMMA under wet and dry conditions. They reported that storage under dry conditions resulted in greater deformation than wet conditions. However, they only compared the distance between two points of the 3D data using a best-fit alignment. This alignment process repeatedly matches the nearest points of the corresponding images in a virtual space on a program. This 3D analysis method enabled the measurement of morphological differences in shapes. They did not examine the deformation area or volume. In their study, the "surface deviation" was defined as the distance between two points of 3D data from this best-fit alignment. In addition, because denture bases and artificial teeth are made of different materials, it is not clear how they deform in three dimensions in an intraoral device fabricated from a single material, such as OcA. Some studies [12,13] have evaluated SB based on three-dimensional measurements of the occlusal wear of OcA. However, the three-dimensional morphological deformation of OcA due to different storage methods has not been examined. Therefore, this study provides basic data for evaluating the three-dimensional morphological deformation of OcA.

In recent years, non-contact 3D scanners have been increasingly used. Compared with contact 3D scanners, which are conventionally considered highly accurate, non-contact 3D

scanners are widely used in the dental field because of their simplicity, quick measurement time, and non-inferior accuracy [14].

This study used a dental model scanner to evaluate OcA deformation under dry and wet storage conditions.

## Material and methods

### Materials

Impressions were made from the upper and lower parts of a dental model (E1-500A-U/ 500A-L; NISSIN, Kyoto, Japan), as shown in Fig 1, using ready-made trays (Human Tray; Maruichi, Tokushima, Japan) and alginate impression materials (ALGINoplast: Kulzer, Hanau, Germany). Plaster models were made using a hard plaster (Zostone dental hard plaster; Shimomura Gypsum Co., Saitama, Japan). The plaster model was mounted on a Gysi Simplex OU-II articulator, and the incisor guidance needle was adjusted to elevate 1.5 mm on the first molar. As grinding at the SB was performed over the canine edge and/or molar cusp [15], the incisal and buccal cusps were set horizontally 1 mm beyond the cusp, the palatal thickness of the OcA was set at 2 mm, and its length was 10 mm above the tooth neck [12] (Fig 2).

The wax mold prepared on the working model was embedded in a metal flask, flowing wax, and then a heat-polymerized resin (Acron: GC) was used at a mix ratio of 0.43 mL/g, as per the manufacturer's instructions. The flask was test-pressure-deburred thrice at approximately 40 kgf/cm$^3$ and then heat-polymerized in hot water at 70°C for 8 h, followed by slow cooling at room temperature. OcA was given occlusal contacts in all dentition rows and group functions in canines and premolars and polished. Ceramic spheres (ZrO$_2$ spheres; Sato Iron Works, Osaka, Japan; JIS standard S28 grade, 2 mm in diameter) were embedded in the equivalent area of the 63⊥36 centrifugal corners of the OcA (Fig 3).

After polishing, OcA was stored in water for one month to reduce the effects of water-absorption-induced expansion and polymerization shrinkage of PMMA and to hydrate it. After one month of storage in water, the product was stored under wet and dry conditions for four weeks. Water was placed in a zippered polyethylene bag for wet storage and stored in a dedicated storage case with the bag sealed. The samples were stored in a special case to prevent contact with water for dry storage. Both storage conditions were maintained at room temperature (22.0–24.0°C).

### Analysis method

OcA 3D measurements were performed using a noncontact 3D scanner (Identica, MEDIT, Seoul, Korea). The OcA surface was sprayed with an optical impression aid for dental laboratory use (Angel Scan Spray; DENTACO, Essen, Germany) and mounted on a jig for photography before measurements were taken.

The accuracy tests were conducted in advance using Identica. As an accuracy test, four measurements were performed using the Identica for each of three transparent-colored acrylic resin spheres with radii of 2.5/8 inches (7,940 μm, Sato Tekko, Toyama, Japan) mounted on a plaster base as shown in Fig 4 (N = 12).

From the obtained stereolithography (STL) data, 100 points were randomly selected from the sphere surface, and the sphere radius was calculated 1,000 times using the least-squares method to calculate the mean value (accuracy) and standard deviation (precision) of each sphere [14]. A difference of 10.7 μm between the true and measured values of the sphere radius and a standard deviation of 27.9 μm was observed.

The wet-conditioned storage group is referred to as W and the dry-conditioned storage group is denoted as D. As shown in Fig 5, OcA was measured on day 0 after one month of

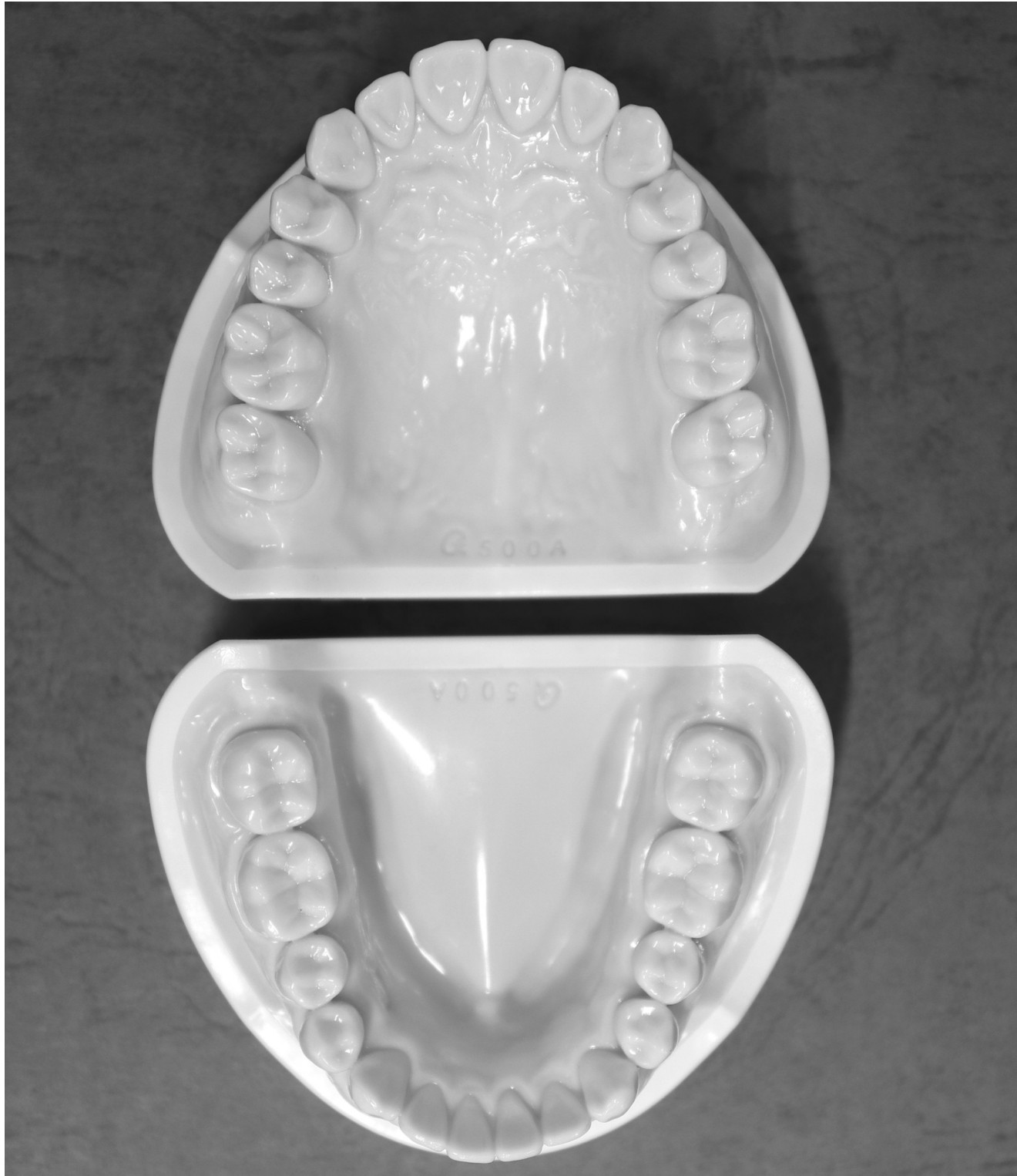

**Fig 1. The upper and lower of a dental model.**

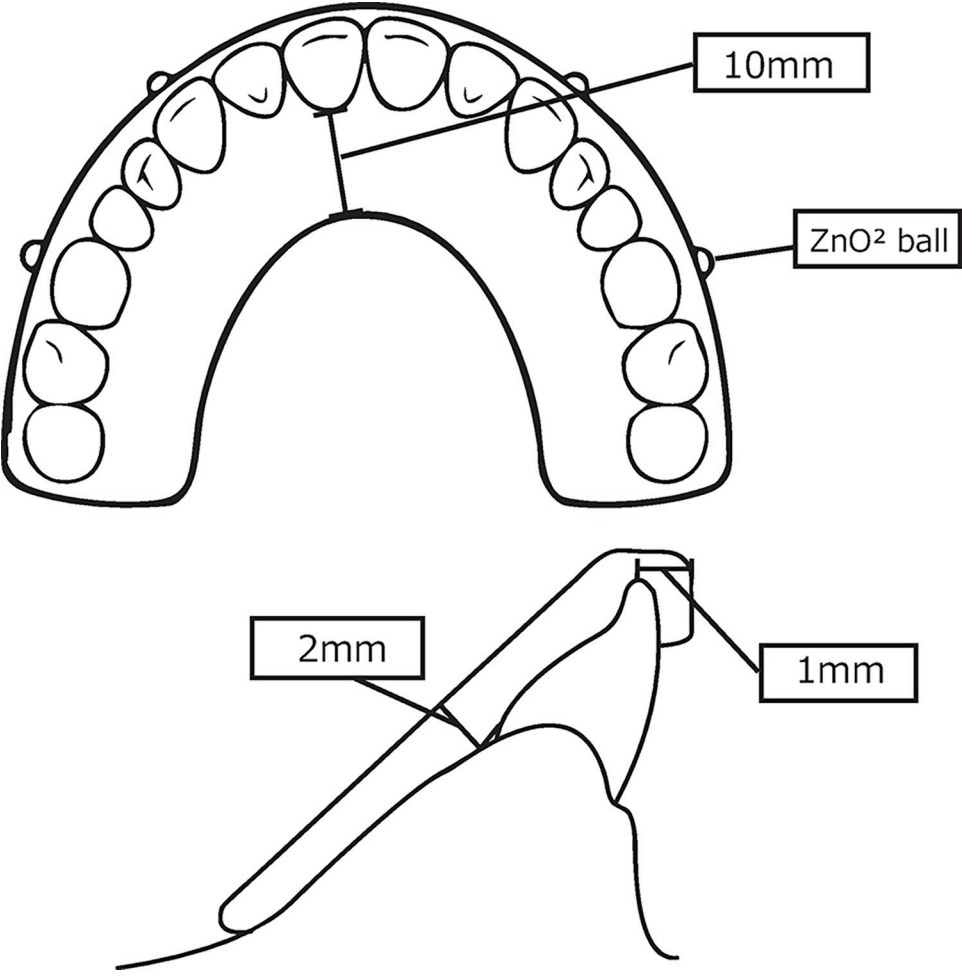

**Fig 2. Design drawing of the OcA.**

hydration (W0) after 4 weeks of storage under wet conditions (day 28: W4, D0), and after 4 weeks of storage under dry conditions (day 56: D4). To avoid inter-rater errors, OcA measurements were performed by a single researcher.

The 3D data analysis uses a best-fit alignment, which is a process that repeatedly matches the nearest points of the corresponding images in a virtual space on a program and calculates the difference between the two images. Using the 3D measurement software, ZEISS Inspect (ZEISS, Oberkochen, Germany), the best-fit alignment of W0 and W4 (day 28) was performed under wet conditions using the STL data for W0 (day 0) as the reference. For the dry conditions, the STL data for D0 (day 28) were used as the reference, and a best-fit alignment of D0 and D4 (day 56) was used to compare the wet and dry conditions. The same test samples were used for comparison.

The surface deviation (mm) was calculated from a best-fit alignment of the STL data for the entire OcA using ZEISS Inspect, with the quantities above the base data in the + direction and those below in the—direction. Since the accuracy test results showed that the error from the true value was within 40 μm, ± 40 μm was used as the cutoff value to calculate the deformation in the—direction below -40 μm and in the + direction above 40 μm.

The area was calculated from the corresponding coordinate plane to the surface deviation in the ± direction considered as the amount of deformation, and this was considered as the

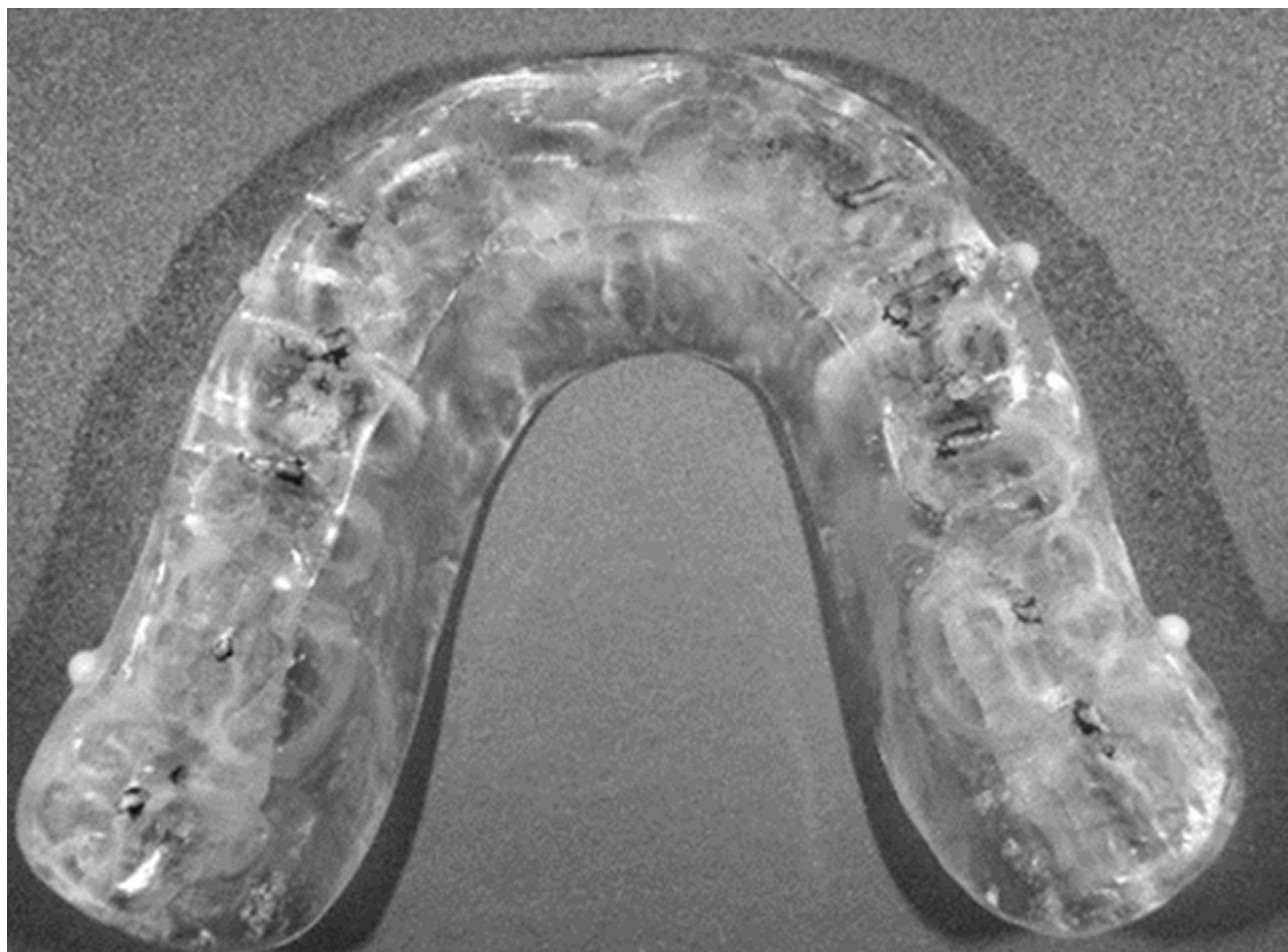

**Fig 3. The OcA occlusal.**

surface area of deformation ($mm^2$). The volume was calculated from the surface area and surface deviation and was used as the deformed volume. From these data, the maximum amount of deviation in the ± direction and the deformed area and volume in the ± direction were used as outcomes.

## Statistical analysis

EZR version 1.68 (Jichi Medical University Saitama Medical Center, Saitama, Japan) [16] was used for the statistical analysis. The maximum deviation in the ± direction, the deformed area in the ± direction, and the volume of the wet condition group (W group) and the dry condition group (D group) were compared using Wilcoxon's signed rank test. The significance level was set at $P < 0.05$.

## Results

A typical example color map of the data superimposed on W0 and W4 for OcA in the W group and D0 and D4 for OcA in the D group is shown (Fig 6).

Deformations in the positive direction are mapped in red, those with no significant deformations are mapped in green, and those in the negative direction are mapped in blue. Visual

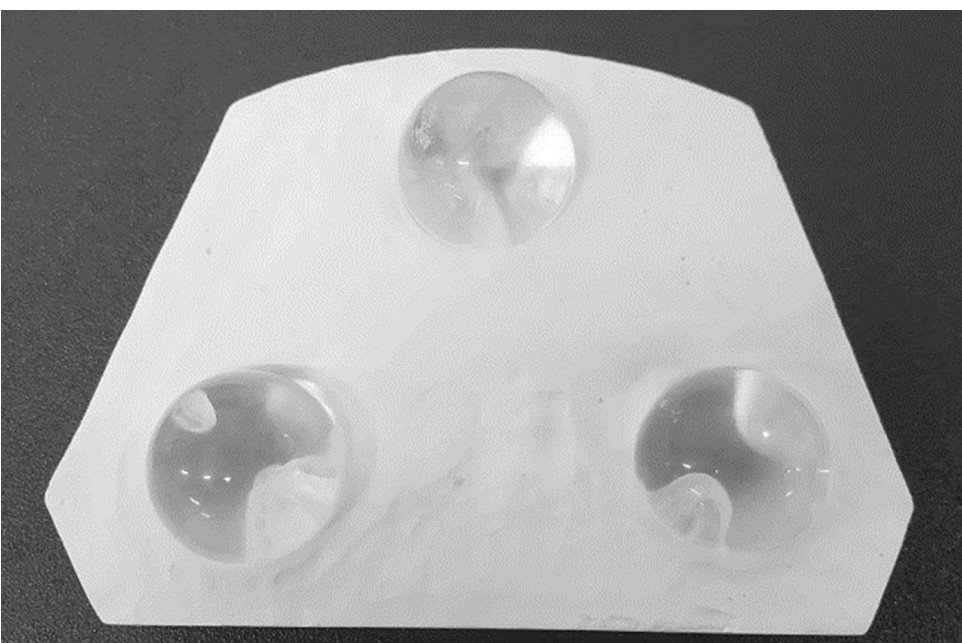

**Fig 4. Plaster base with transparent colored acrylic resin balls.**

evaluation of the 3D deformation of the OcA showed no significant deformation over time in the W group; however, deformation patterns were detected over time in the D group. The posterior limb showed deformation in the direction (blue), and the occlusal surface showed deformation in the + direction (red) toward the anterior limb and in the direction (blue) toward the posterior limb (Fig 6). A typical example of fault deviation showing a cross-sectional deviation in a molar from the D group x-axis plane is shown (Fig 7).

The cross-sectional deviation is the difference in the surface shape between superimposed polygon data in an arbitrary cross-section of the polygon data. Fig 6 also shows + deformation in the forward direction and − deformation in the backward direction around the peak of the step in the guide. A typical example of cross-sectional deviation from the D-group Y-axis plane is shown in Fig 8.

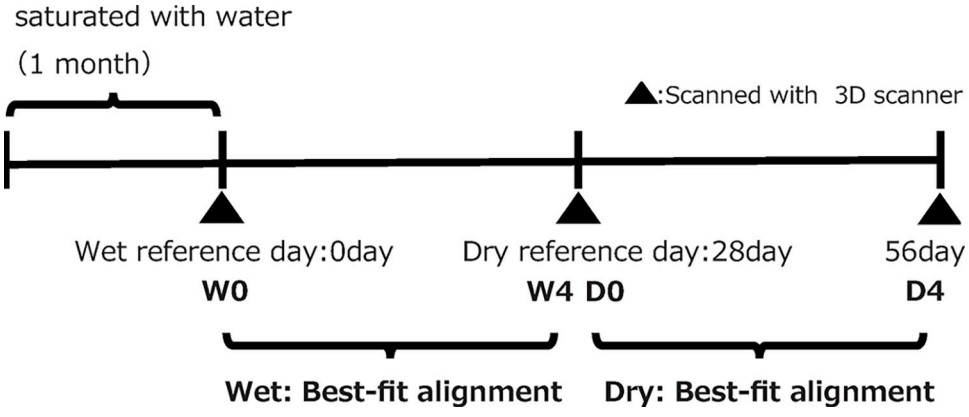

**Fig 5. Research schedule.**

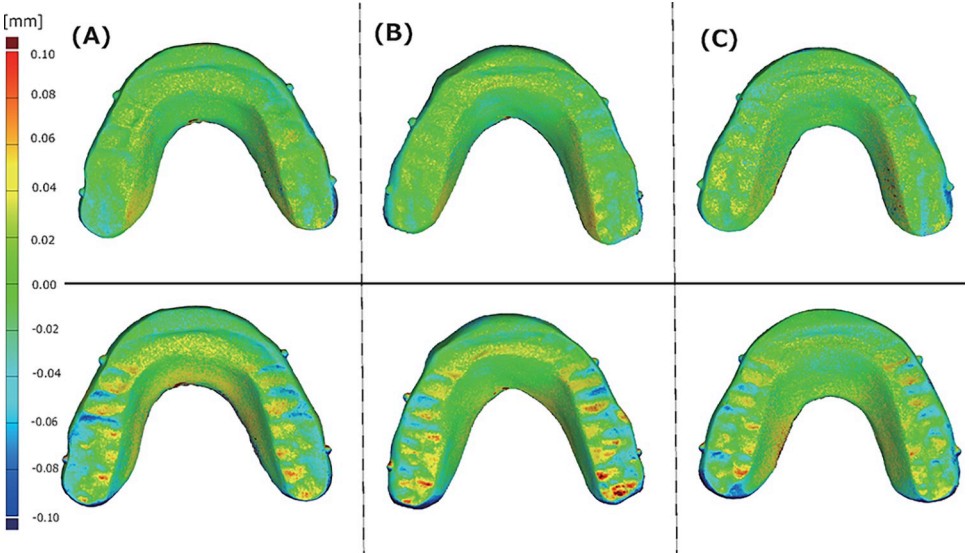

**Fig 6.** Sample (A), (B), and (C) represent typical OcA deformations after 4 weeks of wet and dry storage, with the top row displaying a color map evaluation under wet conditions and the bottom row under dry conditions.

Deformation was observed in a + direction toward the center and a − direction toward the outside. This characteristic deformation was observed in all eight OcAs in Group D.

The deformations of OcAs in the W and D groups and the results of the Wilcoxon signed-rank sum test are shown in Fig 9.

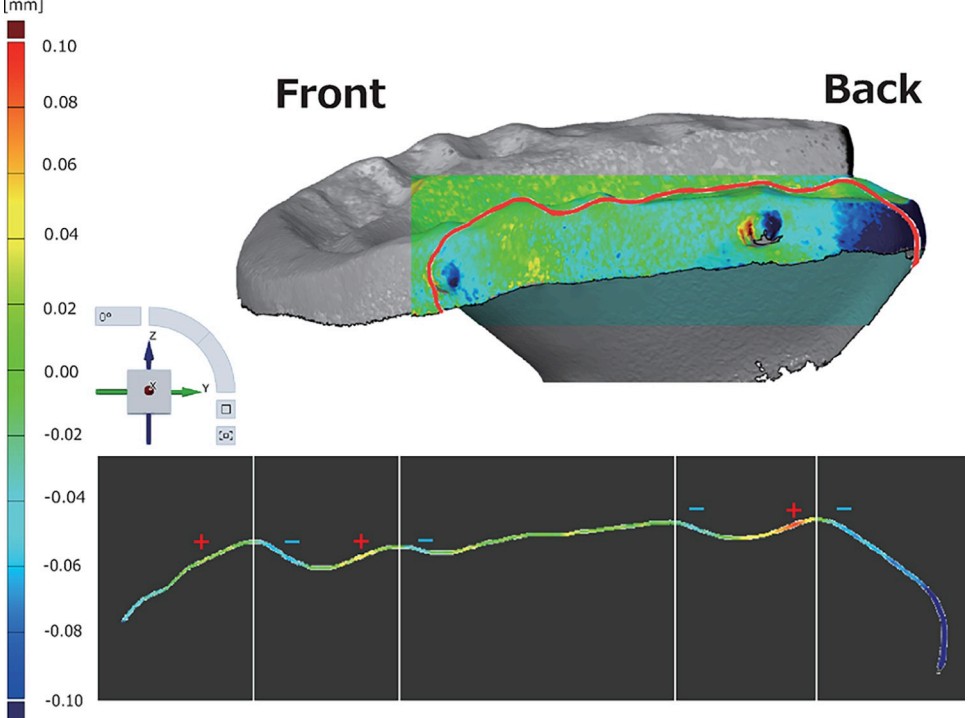

**Fig 7. Cross-sectional deviation evaluating the sagittal section from the X-axis view shows the deviation in the forward + direction and backward—direction (upper).** The lateral view of the OcA and the sagittal section are indicated by red lines (lower).

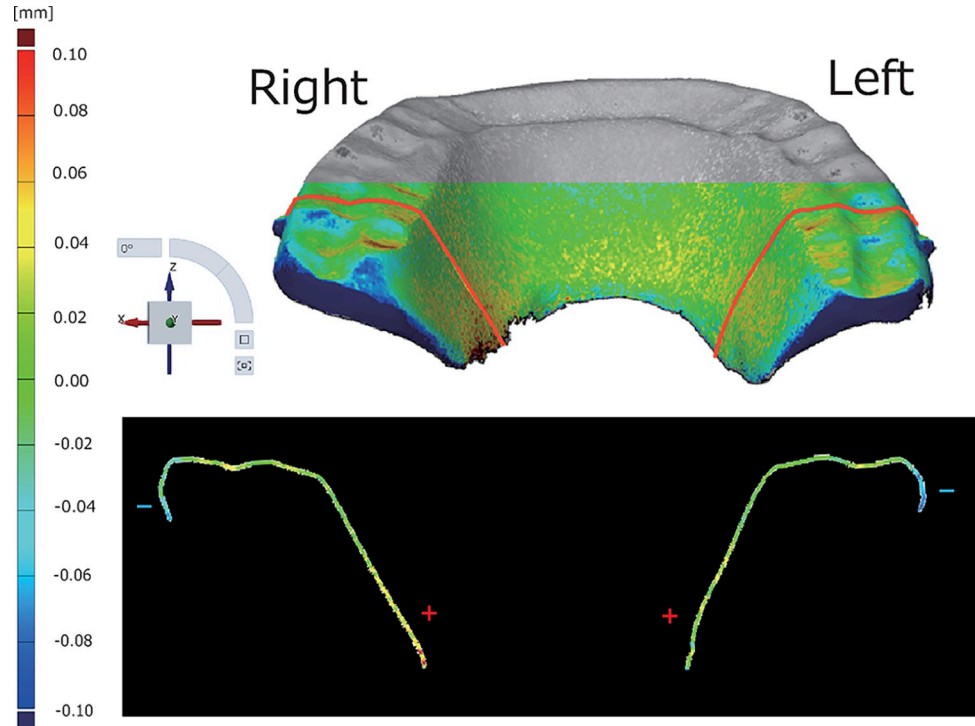

**Fig 8. Cross-sectional deviation evaluating the frontal section from the Y-axis view shows the deviation in the— direction toward the buccal and the + direction toward the palatal (upper) region.** The posterior view of the OcA and the frontal section are indicated by red lines (lower).

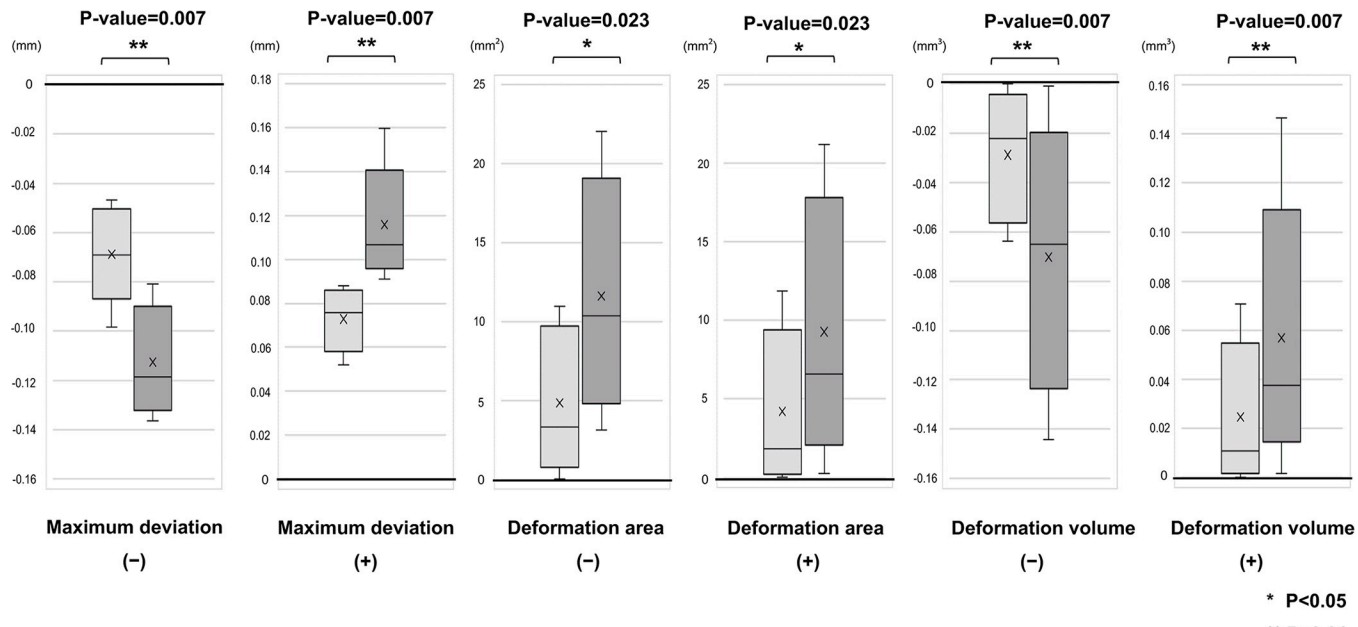

**Fig 9. Comparison of deformation outcomes in the ± direction under wet and dry storage conditions (*P < 0.05, **P < 0.01).** From left to right, the box-and-whisker plots the maximum deviation in the—direction, maximum deviation in the + direction, deformation area in the—direction, deformation area in the + direction, deformation volume in the—direction, and deformation volume in the + direction.

The median deviations were 0.069 mm in the—direction, 0.075 mm in the + direction in the W group, 0.118 mm in the—direction, and 0.106 mm in the + direction in the D group.

The surface areas of the W group were 3.898 mm$^2$ and 1.836 mm$^2$ in the and + directions, respectively, whereas those of the D group were 10.351 mm$^2$ and 6.612 mm$^2$ in the and + directions, respectively.

The volume of deformation was 0.221 mm$^3$ in the—direction and 0.107 mm$^3$ in the + direction in the W group while 0.649 mm$^3$ in the—direction, and 0.376 mm$^3$ in the + direction in the D group.

All outcomes showing these deformities differed significantly between groups D and W ($P < 0.05$).

## Discussion

### Material properties

OcAs are often used for long periods because of their role in distributing excessive occlusal forces, reducing the burden on the TMJ and oral cavity, and protecting the TMJ system from damage to the dentition and periodontal tissues due to pathological occlusal wear, compromised bite, and TMJ arthritis. Therefore, after OcA is precisely fabricated, it is important to minimize its deformation and maintain its morphology to allow it to function.

In this study, the surface deviation, surface area, and deformed volume were significantly greater in the D group than in the W group. However, the W group also exhibited slight deformation. The deformed volume was approximately 2–3 times greater in group D than in group W.

In a previous study, Izumi [12] performed 3D measurements of the OcA by placing a Co-Cr framework in a heat-cured resin (Acron Clear; GC, Tokyo, Japan) and building an immediate-curing resin (Unifast II Clear, GC) on the occlusal surface. Hirai [13] used OcA, which is made by building a faceted resin (GC) on a 0.75-mm polyester sheet (DURAN, Iserlohn, Germany). In the present study, OcA fabricated from a single material made of PMMA, which is commonly used in clinical practice, was employed to investigate deformation due to storage conditions. PMMA has many advantages, including sufficient mechanical strength, aesthetics, low toxicity, ease of repair, and a simple curing procedure; thus, it has been most used in the manufacture of denture bases since its development in 1945 [17] and as a material for OcA. PMMA is considered to have relatively low deformation after fabrication and excellent color stability. However, none of the molding methods are free from dimensional changes because of polymerization-induced shrinkage, thermal shrinkage, stress deformation, or water absorption. A study on dimensional changes in dentures fabricated with PMMA reported that the greatest dimensional changes occurred in the first month, with no significant changes occurring after two months [18,19]. Therefore, in this study, to suppress dimensional deformation due to polymerization shrinkage and water absorption, the resin was kept in water for 1 month after fabrication to hydrate the resin and perform measurements on the OcA [9].

Recently, 3D-printed resins have also been used as OcA materials. While no specific reports on OcA exist, studies have evaluated the dimensional stability of 3D-printed resins for dentures and crowns. One study reported an average deformation of 0.201 ± 0.055 mm in 3D-printed denture bases after 28 days of storage under dry conditions without direct sunlight [20]. Another study evaluating water absorption and solubility in 3D-printed crowns found that 3D-printed PMMA resin exhibited higher water absorption than polycarbonate resin but lower than heat-polymerized PMMA resin [21]. It has been suggested that 3D-printed materials are constructed in layers, allowing water to penetrate these layers and cause polymer chain movements, resulting in dimensional changes. The presence of free monomers in 3D-printed materials, owing to their low polymerization, has increased water absorption [22]. The

material used in this study absorbed more water than the 3D-printed resin. Therefore, the deformation effect due to drying was considered greater in the heat-polymerized PMMA resin than in the 3D-printed resin. Further investigation is required to evaluate the deformation of 3D-printed OcA over time under different storage conditions.

## Analysis methodology

A spray-type coating was adopted as the optical impression-taking aid material for dental laboratory use because its measurement accuracy has previously been determined, and the thickness irregularity during coating is less for the spray type than for the powder type. However, because the spray type also has a 3 μm effect on the thickness of the measured object, it may be better to use a colored object to improve accuracy without the need to use a spray. To set the cutoff value, Izumi [12] measured the OcAs multiple times before and after use and set the cutoff value at 30 μm from the 95% confidence interval of the measured values. In addition, Hirai [13] measured OcAs twice in three dimensions and, based on the error between the two measurements, assumed that a deformation of 40 μm or less was within the error margin. In the present study, optical impression-taking aids for dental laboratory use were used along with scan spray, and the mean value (accuracy) and standard deviation (precision) of the measurement values were calculated from the algorithm for accuracy testing and set a cutoff value of 40 μm for these accuracy tests [7].

## Clinical applications

The results of this study indicate that OcAs stored under wet conditions were less deformed than those stored under dry conditions, which is consistent with previous reports [10].

In a previous study [11] in which 3D measurements were performed on dentures fabricated with heat-cured resin (PMMA) under dry and wet conditions, the upper complete dentures showed significantly greater deformation under dry conditions than under wet conditions at 2 weeks. In lower complete dentures, deformation due to drying conditions was observed after 4 weeks of preservation, but no significant difference in deformation between conditions was observed. This difference is suspected to be due to the larger surface area of the lower complete denture than that of the maxillary complete denture, which absorbs sufficient water before deflasking. Because both conditions in our study were kept in water for 1 month after deflasking, we think that significant differences were observed between the two conditions.

In the present study, deformations of up to 0.069 mm in the direction and 0.075 mm in the + direction were also observed in the 4-week underwater storage group after one month of hydration.

Based on the diffusion coefficient of water, Braden [23] calculated that a 2-mm-thick denture (heat-cured resin) would hydrate in 200 h when stored in water at body temperature (37.5°C) and would take three times longer to hydrate at room temperature (22.5°C) compared to storage at 37.5°C. In other words, it is saturated after approximately 25 days of storage in water at room temperature at a thickness of 2 mm. In the present study, the OcA had a thickness of approximately 3 mm in the occlusal area of the central incisor, and it is possible that the hydration was insufficient in these areas.

Based on the abovementioned findings, the hydration period may be affected by the thickness of OcA and the storage temperature. Considering the effects of polymerization-induced shrinkage, water absorption, and OcA expansion, we emphasize that increased maintenance and recall are required, particularly during the early stages of use. Considering that the OcA is a periodontal ligament support device that covers the dentition rather than a mucosal support device such as a denture, deformation is more easily perceived by the user, and discomfort is

likely to occur. Previous studies reported a deformation of approximately 2.38 mm in the edentulous maxilla as a deformation of the masticatory mucosa and approximately 2 mm as an acceptable denture deformation [24]. In contrast, Picton et al. [25] reported that normal tooth movement during function is within approximately 100 μm in three dimensions; thus, OcA deformation in clinical practice should remain within this range. This study observed maximum deformations of 75 μm in the W group and 118 μm in the D group as ± direction deviations. Therefore, OcA deformation under dry conditions is considered unacceptable.

Deformation was observed under dry conditions in the posterior margin, palate, and buccal areas of the molars. These areas shrank toward the proximal region, and the degree of positive deformation was greater on the centrifugal palate side than on the central palate side. These results may serve as a basis for patient guidance materials explaining how to store OcAs. Patients could visually understand from these data that if they did not wear the OcA for a long period, they would become deformed and ill-fitted because of the storage conditions. In addition, when dentists adjust the OcA, it is possible to predict the deformity site and the adjustment required to correct the fit. Occlusal surfaces may require occlusal adjustment due to shrinkage toward the front and adjustment during difficult removal due to shrinkage in the form of lifting toward the center.

The mechanism by which OcAs suppress SB is currently unknown; however, this suppression is short-lived. Previous studies reported three-dimensional (3D) analyses of OcA deformation caused by SB [12,13]. This study investigated the deformation of OcA under different storage conditions, which may help in the analysis of SB-induced deformation of OcAs.

## Limitations

This study had three limitations. First, to compare each condition using the best-fit alignment, wet and dry conditions were analyzed using the same test sample. However, the deformation under wet conditions may also be influenced by the carryover effect. Because the greatest dimensional changes due to water absorption and expansion occurred during the first month, with no significant changes observed after two months [18,19], a 1-month water absorption period was adopted in this study. Nevertheless, deformation was also observed under wet conditions, suggesting that different deformation patterns might have been observed if the same period had been evaluated under dry conditions. However, creating test samples with identical OcA morphologies under both dry and wet conditions is challenging. Therefore, this protocol enabled a corresponding group comparison using test samples with identical OcA morphologies under dry and wet conditions.

Second, we adopted the best fit for the entire OcA as the designated range criterion for superimposition. Hirai [13] obtained impressions of the OcA placed on a plaster model and performed 3D measurements. This was because the area of the model palate that was not deformed was used as the reference designation range. However, in this case, deformation may have occurred because of OcA mounting on the model. In this study, only OcA was measured, and a ceramic sphere was installed as a reference point. However, the ceramic sphere moved because of the OcA deformation. However, by focusing on this ceramic sphere, the deformation at the back and forth of the OcA became visually apparent. Previous studies evaluating the deformation of heat-cured resins employed linear or vertical cross-sectional analysis [26,27]. Most previous studies have used optical microscopes or calipers to measure the distances between certain landmark points; however, these methods are limited in determining the overall deformation because the measurement of two points is simply a linear analysis. We believe that using the best-fit measurements in this study revealed three-dimensional changes.

Finally, because OcA deformation is influenced by the material thickness, changes in thickness caused by occlusion or alignment may alter the deformation tendencies. For example, an increased angle of the lateral incisal path can thicken the OcA, leading to greater deformation in the thicker areas. Varying occlusal conditions, such as maxillary prognathism, mandibular prognathism, and open bite, affect the pairing of the upper and lower dentitions. These differences can result in variations in OcA thickness and the corresponding deformation tendencies.

## Conclusions

OcAs made of heat-polymerized resin (PMMA) showed greater deformation in terms of surface deviation, deformed area, and deformed volume under dry conditions than underwater conditions after 4 weeks of storage. Under dry conditions, deformation occurred such that the center of the product shrank toward the center and the centrifugal portion was lifted toward the palate.

## Supporting information

**S1 Table. Summary of OcA deformation (raw data).**
(XLSX)

## Acknowledgments

We thank Editage and Fabillar Jaime, Jr. Moreno for their editorial work and Masamitsu Oshima, Eri Aoki, and Akari Shimizu for their technical assistance.

## Author Contributions

**Conceptualization:** Aya Ozawa, Yoshitaka Suzuki, Emi Uyama.

**Data curation:** Kenichi Hamada.

**Formal analysis:** Aya Ozawa, Yoshitaka Suzuki.

**Methodology:** Aya Ozawa, Kazuo Okura, Toshinori Okawa, Susumu Abe, Kohei Kamoi, Emi Uyama, Kenichi Hamada, Yoshizo Matuka.

**Software:** Aya Ozawa, Toshinori Okawa.

**Supervision:** Aya Ozawa.

**Visualization:** Toshinori Okawa.

**Writing – original draft:** Aya Ozawa.

**Writing – review & editing:** Yoshitaka Suzuki, Kazuo Okura, Toshinori Okawa, Susumu Abe, Kohei Kamoi, Emi Uyama, Kenichi Hamada, Yoshizo Matuka.

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
