## [Decision Letter · Decision Letter 0]

10 Dec 2024

PONE-D-24-47720Three-dimensional analysis using a dental model scanner: Morphological changes of occlusal appliances used for sleep bruxism under dry and wet conditions.PLOS ONE

Dear Dr. Suzuki,

Thank you for submitting your manuscript to PLOS ONE. After careful consideration, we feel that it has merit but does not fully meet PLOS ONE’s publication criteria as it currently stands. Therefore, we invite you to submit a revised version of the manuscript that addresses the points raised during the review process. Please submit your revised manuscript by Jan 24 2025 11:59PM. If you will need more time than this to complete your revisions, please reply to this message or contact the journal office at plosone@plos.org. Please include the following items when submitting your revised manuscript:A rebuttal letter that responds to each point raised by the academic editor and reviewer(s). You should upload this letter as a separate file labeled 'Response to Reviewers'.A marked-up copy of your manuscript that highlights changes made to the original version. You should upload this as a separate file labeled 'Revised Manuscript with Track Changes'.An unmarked version of your revised paper without tracked changes. You should upload this as a separate file labeled 'Manuscript'.

We look forward to receiving your revised manuscript.

Kind regards,

Carlos Alberto Antunes Viegas, DVM; MSc; PhD

Academic Editor

PLOS ONE

Journal Requirements:

2. Thank you for stating the following in your Competing Interests section: [NO authors have competing interests]. Please complete your Competing Interests on the online submission form to state any Competing Interests. If you have no competing interests, please state "The authors have declared that no competing interests exist.", as detailed online in our guide for authors at http://journals.plos.org/plosone/s/submit-now This information should be included in your cover letter; we will change the online submission form on your behalf.

3. We note that your Data Availability Statement is currently as follows: [All relevant data are within the manuscript and its Supporting Information files.] Please confirm at this time whether or not your submission contains all raw data required to replicate the results of your study. Authors must share the “minimal data set” for their submission. PLOS defines the minimal data set to consist of the data required to replicate all study findings reported in the article, as well as related metadata and methods (https://journals.plos.org/plosone/s/data-availability#loc-minimal-data-set-definition). For example, authors should submit the following data: - The values behind the means, standard deviations and other measures reported; - The values used to build graphs; - The points extracted from images for analysis. Authors do not need to submit their entire data set if only a portion of the data was used in the reported study. If your submission does not contain these data, please either upload them as Supporting Information files or deposit them to a stable, public repository and provide us with the relevant URLs, DOIs, or accession numbers. For a list of recommended repositories, please see https://journals.plos.org/plosone/s/recommended-repositories. If there are ethical or legal restrictions on sharing a de-identified data set, please explain them in detail (e.g., data contain potentially sensitive information, data are owned by a third-party organization, etc.) and who has imposed them (e.g., an ethics committee). Please also provide contact information for a data access committee, ethics committee, or other institutional body to which data requests may be sent. If data are owned by a third party, please indicate how others may request data access.

Reviewers' comments:

Reviewer's Responses to Questions

**Comments to the Author**

1. Is the manuscript technically sound, and do the data support the conclusions?

Reviewer #1: Partly

Reviewer #2: Partly

2. Has the statistical analysis been performed appropriately and rigorously? 

Reviewer #1: I Don't Know

Reviewer #2: Yes

3. Have the authors made all data underlying the findings in their manuscript fully available?

Reviewer #1: No

Reviewer #2: Yes

4. Is the manuscript presented in an intelligible fashion and written in standard English?

Reviewer #1: Yes

Reviewer #2: No

5. Review Comments to the Author

Reviewer #1: The article was well-written and presents a replicable methodology. However, it is important to note that resin undergoes dimensional changes, especially when outside a wet environment, where these changes become significant. While studies on 3D-printed OcA for sleep bruxism are limited, the results consistently reflect this inherent behavior of resins in dry conditions. Raw study data and tables with values that can be compared between groups should be provided, if not disclosed, at least be available to reviewers.

Reviewer #2: This research presents the deformation patterns of OcA due to differences in preservation conditions, and this is considered to be an interesting study. However, there are some points that are insufficiently described, so please revise the following points.

1. Abstract

Is “best-fit alignment” widely used? As there are readers from many fields, it is better to use general terms in the abstract.

In addition, does the “best-fit superposition” in the Methods section of the main text indicate the same method? Please provide a specific explanation of the Method section.

2. Material and Methods

・There is no information on the occlusion or alignment of the dentition model you used, it is necessary to add a description or a photograph.

Moreover, do the deformations differ depending on occlusion and alignment? I suggest adding a description to the limitation.

・p.5 l.4 “W-28”, l.6”D-28”

In Figure 3, they are W4 and D4 respectively, so it would be better to change

the description. Lines 1 and 2 of the Result section are also the same.

・You need to add information about how many people were involved in the

measurement and about the inter-rater error.

3. Results

・Please describe what the figures A, B, and C in Figure 4 and the figures in the top and bottom rows show. Also, please show what the legend for the graph in Figure 7 shows.

4.Discussion

・This research focuses on the accuracy of OcA deformation, but how much deformation is clinically acceptable when compared to dentures?

・As was mentioned in the Limitation section, why were the W and D groups

not measured separately? Does the change in the W group not affect the change in the D group?

6. PLOS authors have the option to publish the peer review history of their article (what does this mean?). If published, this will include your full peer review and any attached files.

Reviewer #1: No

Reviewer #2: No

---

## [Author Response · Author response to Decision Letter 0]

1 Jan 2025

Journal Requirements

1. Please ensure that your manuscript meets PLOS ONE's style requirements, including those for file naming. The PLOS ONE style templates can be found at https://journals.plos.org/plosone/s/file?id=wjVg/PLOSOne_formatting_sample_main_body.pdf

>Thank you. We have updated the file name and format size to align with PLOS ONE's style requirements. Re-calibrated by author and Editage.

2. Thank you for stating the following in your Competing Interests section: [NO authors have competing interests]. Please complete your Competing Interests on the online submission form to state any Competing Interests. If you have no competing interests, please state "The authors have declared that no competing interests exist.", as detailed online in our guide for authors at http://journals.plos.org/plosone/s/submit-now This information should be included in your cover letter; we will change the online submission form on your behalf.

Thank you for the suggestion. We have declared that no competing interests exist in the cover letter.

3. We note that your Data Availability Statement is currently as follows: [All relevant data are within the manuscript and its Supporting Information files.] Please confirm at this time whether or not your submission contains all the raw data required to replicate the results of your study. 

Thank you for the suggestion. All raw data is added to "Summary of OcA deformation" as "Supporting Information.” Page 21.

Review Comments

Reviewer #1: The article was well-written and presents a replicable methodology. However, it is important to note that resin undergoes dimensional changes, especially when outside a wet environment, where these changes become significant. While studies on 3D-printed OcA for sleep bruxism are limited, the results consistently reflect this inherent behavior of resins in dry conditions. Raw study data and tables with values that can be compared between groups should be provided, if not disclosed, at least be available to reviewers.

Thank you for the careful review and guidance. In this study, all raw data is added to the "Summary of OcA deformation" as "Supporting Information.” In addition, the study of dimensional stability literature for 3D-printed resins was examined, and the following text was added to the "Discussion.” Page 12, Line 281.

Recently, 3D-printed resins have also been used as OcA materials. While no specific reports on OcA exist, studies have evaluated the dimensional stability of 3D-printed resins for dentures and crowns. One study reported an average deformation of 0.201 ± 0.055 mm in 3D-printed denture bases after 28 days of storage under dry conditions without direct sunlight [20]. Another study evaluating water absorption and solubility in 3D-printed crowns found that 3D-printed PMMA resin exhibited higher water absorption than polycarbonate resin but lower than heat-polymerized PMMA resin [21]. It has been suggested that 3D-printed materials are constructed in layers, allowing water to penetrate these layers and cause polymer chain movements, resulting in dimensional changes. The presence of free monomers in 3D-printed materials, owing to their low polymerization, has increased water absorption [22]. The material used in this study absorbed more water than the 3D-printed resin. Therefore, the deformation effect due to drying was considered greater in the heat-polymerized PMMA resin than in the 3D-printed resin. Further investigation is required to evaluate the deformation of 3D-printed OcA over time under different storage conditions.

Reviewer #2: This research presents the deformation patterns of OcA due to differences in preservation conditions, and this is considered to be an interesting study. However, there are some points that are insufficiently described, so please revise the following points.

Thank you for your careful peer review and comments. Our responses to the points raised are as follows. The italicized text has been added to the revised manuscript. 

1. AbstractIs “best-fit alignment” widely used? As there are readers from many fields, it is better to use general terms in the abstract.

Thank you for pointing this out. We have unified "best-fit superposition" with "best-fit alignment" because "best-fit alignment" is the term generally used in a paper investigating the deformation of OcA[12,13]. We added a description of best-fit alignment to methods in "Abstract" and "Material and Methods.”

Page 3, Line 44

that aligns the nearest points of the corresponding images in a virtual space and calculates the difference between the two images. 

Page 8, Line 169

The 3D data analysis uses a best-fit alignment, which is a process that repeatedly matches the nearest points of the corresponding images in a virtual space on a program and calculates the difference between the two images. 

 Page 8, Lines 169 and 177

and a best-fit alignment of D0 and D4 (day 56) was used to compare the wet and dry conditions. The comparison was performed using the same test sample.

The surface deviation (mm) was calculated from a best-fit alignment of the STL data for the entire OcA using ZEISS Inspect,

2. Material and Methods

・There is no information on the occlusion or alignment of the dentition model you used, it is necessary to add a description or photograph.

Thank you for pointing this out. The occlusal of the OcA is added as Fig. 3 on page 7, and the occlusion and alignment of the model as Fig. 1.

Fig 1. The upper and lower of a dental model

Fig 3. The OcA occlusal

Moreover, do the deformations differ depending on occlusion and alignment? I suggest adding a description to the limitation.

＞As you point out, deformations may vary depending on the occlusal condition and alignment. The following text has been added to the "Limitation.”

Page 17, Line 389

Finally, because OcA deformation is influenced by the material thickness, changes in thickness caused by occlusion or alignment may alter the deformation tendencies. For example, an increased angle of the lateral incisal path can thicken the OcA, leading to greater deformation in the thicker areas. Varying occlusal conditions, such as maxillary prognathism, mandibular prognathism, and open bite, affect the pairing of the upper and lower dentitions. These differences can result in variations in OcA thickness and the corresponding deformation tendencies.

・In Figure 3, they are W4 and D4 respectively, so it would be better to changethe description. Lines 1 and 2 of the Result section are also the same.

Thank you for pointing this out. We agree with the reviewer’s suggestion. We corrected the text to W4 and D4 in the description and "Results.”

Page 8, Line 162

As shown in Fig 5, OcA was measured on day 0 after one month of hydration (W0) after 4 weeks of storage under wet conditions (day 28: W4, D0), and after 4 weeks of storage under dry conditions (day 56: D4). 

Page 8, Line 171

Using the 3D measurement software, ZEISS Inspect (ZEISS, Oberkochen, Germany), the best-fit alignment of W0 and W4 (day 28) was performed under wet conditions using the STL data for W0 (day 0) as the reference. For the dry conditions, the STL data for D0 (day 28) were used as the reference, and a best-fit alignment of D0 and D4 (day 56) was used to compare the wet and dry conditions. The same test samples were used for comparison.

Page 9, Line 198

A typical example color map of the data superimposed on W0 and W4 for OcA in the W group and D0 and D4 for OcA in the D group is shown (Fig 6). 

・You need to add information about how many people were involved in themeasurement and about the inter-rater error.

Thank you for the suggestion. The same examinee performed the measurements to avoid variations in the measurement methods. The following text has been added to the " Materials and Methods section. 

Page 8, Line 164

To avoid inter-rater errors, OcA measurements were performed by a single researcher.

3. Results

・Please describe what the figures A, B, and C in Figure 4 and the figures in the top and bottom rows show. Also, please show what the legend for the graph in Figure 7 shows.

Thank you for pointing this out. Two more figures have been added, and the notation has shifted (Fig. 4 ⇒ Fig. 6, Fig. 7 ⇒ Fig. 9). The following explanatory text has been added to each legend:

Page 9, Line 201

Fig 6. Sample (A), (B), and (C) represent typical OcA deformations after 4 weeks of wet and dry storage, with the top row displaying a color map evaluation under wet conditions and the bottom row under dry conditions.

Page 11, Line 234

Fig 9. Comparison of deformation outcomes in the ± direction under wet and dry storage conditions (*P < 0.05, **P < 0.01). From left to right, the box-and-whisker plots the maximum deviation in the - direction, maximum deviation in the + direction, deformation area in the - direction, deformation area in the + direction, deformation volume in the - direction, and deformation volume in the + direction.

4. Discussion

・This research focuses on the accuracy of OcA deformation, but how much deformation is clinically acceptable when compared to dentures?

Thank you for pointing this out. We believe that your suggestion is clinically necessary. The following text has been added to the "Discussion" section:

 Page 15, Line 339

 Previous studies reported a deformation of approximately 2.38 mm in the edentulous maxilla as a deformation of the masticatory mucosa and approximately 2 mm as an acceptable denture deformation [24]. In contrast, Picton et al. [25] reported that normal tooth movement during function is within approximately 100 μm in three dimensions; thus, OcA deformation in clinical practice should remain within this range. This study observed maximum deformations of 75 μm in the W group and 118 μm in the D group as ± direction deviations. Therefore, OcA deformation under dry conditions is considered unacceptable. 

・As was mentioned in the Limitation section, why were the W and D groups

not measured separately? Does the change in the W group not affect the change in the D group?

Thank you for your valuable comment. As you pointed out, the results of Group D may have been influenced by those of Group W. However, separating the W and D groups would result in comparing test specimens with different morphologies, leading to a non-corresponding group comparison. In this study, we adopted this protocol because we aimed to create test specimens with identical OcA morphologies under both dry and wet conditions and to perform a corresponding group comparison using the same specimens. In the future, we plan to investigate whether the results of this study can be reproduced using the suggested methods.

The following text was added to the "Limitations." 

In-Page 16, Line 364

First, to compare each condition using the best-fit alignment, wet and dry conditions were analyzed using the same test sample. However, the deformation under wet conditions may also be influenced by the carryover effect. Because the greatest dimensional changes due to water absorption and expansion occurred during the first month, with no significant changes observed after two months [18,19], a 1-month water absorption period was adopted in this study. Nevertheless, deformation was also observed under wet conditions, suggesting that different deformation patterns might have been observed if the same period had been evaluated under dry conditions. However, creating test samples with identical OcA morphologies under both dry and wet conditions is challenging. Therefore, this protocol enabled a corresponding group comparison using test samples with identical OcA morphologies under dry and wet conditions.

---

## [Decision Letter · Decision Letter 1]

19 Jan 2025

Three-dimensional analysis using a dental model scanner: Morphological changes of occlusal appliances used for sleep bruxism under dry and wet conditions.

PONE-D-24-47720R1

Dear Dr. Yoshitaka Suzuki,

We’re pleased to inform you that your manuscript has been judged scientifically suitable for publication and will be formally accepted for publication once it meets all outstanding technical requirements.

Kind regards,

Carlos Alberto Antunes Viegas, DVM; MSc; PhD

Academic Editor

PLOS ONE

Additional Editor Comments (optional):

Reviewers' comments:

Reviewer's Responses to Questions

**Comments to the Author**

1. If the authors have adequately addressed your comments raised in a previous round of review and you feel that this manuscript is now acceptable for publication, you may indicate that here to bypass the “Comments to the Author” section, enter your conflict of interest statement in the “Confidential to Editor” section, and submit your "Accept" recommendation.

Reviewer #1: All comments have been addressed

Reviewer #2: All comments have been addressed

2. Is the manuscript technically sound, and do the data support the conclusions?

Reviewer #1: Yes

Reviewer #2: Yes

3. Has the statistical analysis been performed appropriately and rigorously? 

Reviewer #1: I Don't Know

Reviewer #2: Yes

4. Have the authors made all data underlying the findings in their manuscript fully available?

Reviewer #1: Yes

Reviewer #2: Yes

5. Is the manuscript presented in an intelligible fashion and written in standard English?

Reviewer #1: Yes

Reviewer #2: Yes

6. Review Comments to the Author

Reviewer #1: (No Response)

Reviewer #2: (No Response)

7. PLOS authors have the option to publish the peer review history of their article (what does this mean?). If published, this will include your full peer review and any attached files.

Reviewer #1: No

Reviewer #2: No

---

## [Editor Report · Acceptance letter]

23 Jan 2025

PONE-D-24-47720R1 

PLOS ONE

Dear Dr. Suzuki, 

I'm pleased to inform you that your manuscript has been deemed suitable for publication in PLOS ONE. Congratulations! Your manuscript is now being handed over to our production team.

Kind regards, 

on behalf of

Dr. Carlos Alberto Antunes Viegas 

Academic Editor

PLOS ONE